# An Integrative Transcriptomics and Proteomics Approach to Identify Putative Genes Underlying Fruit Ripening in Tomato near Isogenic Lines with Long Shelf Life

**DOI:** 10.3390/plants12152812

**Published:** 2023-07-29

**Authors:** Melisa Di Giacomo, Tatiana Alejandra Vega, Vladimir Cambiaso, Liliana Amelia Picardi, Gustavo Rubén Rodríguez, Javier Hernán Pereira da Costa

**Affiliations:** 1Instituto de Investigaciones en Ciencias Agrarias de Rosario (IICAR-CONICET-UNR), Campo Experimental Villarino, Facultad de Ciencias Agrarias, Universidad Nacional de Rosario, Zavalla S2125ZAA, Santa Fe, Argentina; digiacomo@iicar-conicet.gob.ar (M.D.G.); vega@iicar-conicet.gob.ar (T.A.V.); cambiaso@iicar-conicet.gob.ar (V.C.); grodrig@unr.edu.ar (G.R.R.); 2Cátedra de Genética, Facultad de Ciencias Agrarias, Universidad Nacional de Rosario, Zavalla S2125ZAA, Santa Fe, Argentina; lpicardi@unr.edu.ar

**Keywords:** *S*. *lycopersicum*, differential expression, ethylene response factor, fruit softening, cell wall remodeling

## Abstract

The elucidation of the ripening pathways of climacteric fruits helps to reduce postharvest losses and improve fruit quality. Here, we report an integrative study on tomato ripening for two near-isogenic lines (NIL115 and NIL080) with *Solanum pimpinellifolium* LA0722 introgressions. A comprehensive analysis using phenotyping, molecular, transcript, and protein data were performed. Both NILs show improved fruit firmness and NIL115 also has longer shelf life compared to the cultivated parent. NIL115 differentially expressed a transcript from the APETALA2 ethylene response transcription factor family (AP2/ERF) with a potential role in fruit ripening. E4, another ERF, showed an upregulated expression in NIL115 as well as in the wild parent, and it was located physically close to a wild introgression. Other proteins whose expression levels changed significantly during ripening were identified, including an ethylene biosynthetic enzyme (ACO3) and a pectate lyase (PL) in NIL115, and an alpha-1,4 glucan phosphorylase (Pho1a) in NIL080. In this study, we provide insights into the effects of several genes underlying tomato ripening with potential impact on fruit shelf life. Data integration contributed to unraveling ripening-related genes, providing opportunities for assisted breeding.

## 1. Introduction

Tomato is widely used as a model system to study the regulatory network underlying fruit ripening [1]. Several biochemical processes, such as phytohormone signaling pathways, cell wall modification, pigment accumulation, and transcription factor networks were studied during ripening [2,3,4]. In tomato, there are three well-known ripening mutations: *rin*, *nor,* and *Cnr* [5,6,7]. These are commercially important mutants that extend shelf life (SL), but also produce undesirable effects on fruit quality. Other key ripening factors are softening-associated metabolic enzymes, such as polygalacturonase, pectate lyase, and pectin methylesterase [8,9,10]. Silencing each of these genes produces only minimal progress in delaying fruit ripening. These results demonstrate the complexity of this trait and its polygenic inheritance.

Recent advances in the field of omics accelerated in-depth studies of molecular mechanisms [11,12] and the integration of different omics approaches helped to elucidate ripening pathways. For instance, a combined transcript, protein, and metabolite analysis of ripening mutants revealed multiple ethylene-associated events during tomato ripening [13], and transcriptome and methylome analysis showed the effects of ripening on and off the vine on tomato flavor [14]. Furthermore, the effect of the low-temperature storage of green tomatoes on gene and protein expression levels and ethylene response was studied [15]. Proteomics and metabolomics identified several compounds related to carbohydrates, amino acids, and fatty acid metabolisms at different ripening stages and salt treatments [16]. Bertero et al. [17] proposed a consensus chaperone network during tomato fruit ripening based on an in silico omics approach. Additionally, a recent study integrated transcriptome and epigenome changes, providing new molecular insights underlying long SL tomato fruits [18]. Several authors recently published transcriptomic and quantitative proteomic studies on tomatoes in relationship to fruit quality and ripening [19,20,21,22,23,24]. These studies contributed to a better understanding of the molecular mechanisms and metabolic pathways involved in tomato ripening.

The elucidation of ripening-related genes provides new tools for tomato breeding that could help to increase SL and improve fruit quality. The development of tomato varieties with increased fruit SL and quality will make a great impact by reducing food waste, with positive implications to consumers’ perception. Wild relative germplasm represents an important reservoir of alleles to improve these traits [25,26,27]. Previously, we demonstrated that the *S. pimpinellifolium* accession LA0722 carries long SL genes that also improved fruit quality [28,29,30,31]. Thus, we developed a collection of near-isogenic lines (NILs) with introgressions from LA0722 [32]. These wild introgressions improved fruit quality when compared to the recurrent parent Caimanta (CAI) of *S*. *lycopersicum*. From this collection, specific NILs displayed longer SL and reduced fruit softening, converting them into valuable genetic resources to study the components that define these traits and to unveil the genetic bases of fruit ripening.

The integration of different high-throughput approaches and the usefulness of long SL tomato lines could help to elucidate novel players in ripening pathways. Therefore, our aim was to gather additional evidence of tomato fruit ripening by assessing two long SL NILs with *S*. *pimpinellifolium* wild introgressions. In order to identify key ripening genes, we performed a comprehensive analysis at different molecular levels, including genomic, transcriptomic, and proteomic approaches. The combination of tools and the data integration contributed to the elucidation of molecular mechanisms underlying physiological processes leading to long SL and reduced fruit softening in the NILs.

## 2. Results

### 2.1. Phenotypic Analysis

The parental genotypes (CAI and LA0722) of NIL115 and NIL080 (Appendix A) presented significant differences (*p*-value < 0.05) for all traits except for color index a/b and pH (Figure 1 and Appendix A). CAI was characterized by higher values in size traits such as diameter (D), height (H), and weight (W). LA0722 had smaller (0.74 ± 0.04 g of W) round fruits (shape (Sh, H/D ratio) = 0.93 ± 0.01), while CAI showed heavier (76.89 ± 7.47 g of W) flattened fruits (Sh = 0.72 ± 0.01). LA0722 showed increased values for quality traits such as SL (15.19 ± 3.07 days), firmness (F = 57.33 ± 1.53), soluble solids content (SSC = 9.10 ± 0.65 °Brix), and titratable acidity (TA = 0.96 ± 0.14 g citric and malic acid/100 g of homogenized juice).

Regarding the derived NILs, both NILs showed no difference for W compared to CAI. NIL080 also presented a similar size and morphology (Sh and locule number (LN)). Both NILs exhibited fleshy fruits (0.67 ± 0.09 cm and 0.47 ± 0.02 cm of pericarp thickness (PT) for NIL115 and NIL080, respectively), also with no differences compared to CAI. As for SL, NIL115 showed differences with CAI (4.50 ± 1.61 days for CAI vs. 14.00 ± 1.38 days for NIL115). On the other hand, both NILs presented significantly firmer fruits than CAI (Figure 1). For the a/b color index, NIL080 exhibited a mean negative value and was different to CAI.

### 2.2. Molecular Characterization of the NILs

A different number of wild introgressions were revealed in the NILs by the molecular characterization (Figure 2). On average, NIL115 had eight introgressions, while NIL080 only had four. The lower number of wild introgressions in NIL080 was expected, as this NIL has an additional round of backcrossing. A total of 3.93% of the NIL115 genome presented homozygous wild introgressions (PP), while a 4.31% were heterozygous (CP). In NIL080, 2.36% of the genome was PP and 1.57% was CP.

Target segments determined by the simple sequence repeat (SSR) markers comprehended adjacent insertion/deletion (InDel) markers that could not recombine due to their proximity, indicating the presence of larger wild segments. Adjacent to the SSR115 (SL4.0ch05:2,779,337), wild introgressions were found on the IND5-0325 (SL4.0ch05:3,300,671) and the IND5-0697 (SL4.0ch05:7,037,638) for NIL115. Near SSR080 (SL4.0ch11:2,309,947), the IND11-0017 (SL4.0ch11:186,598) was segregating in NIL080 individuals. Both NILs also had off-target introgressions localized in other chromosomes. NIL115 presented two wild introgressions on the bottom of chromosome 3 (IND3-5470, SL4.0ch03:49,258,124 and FW3.2, SL4.0ch03:59,219,937), and one on each of the following chromosomes: 8 (IND8-6582, SL4.0ch08:63,974,144), 10 (IND10-0429, SL4.0ch10:4,087,865), and 11 (FAS, SL4.0ch11:52,946,405). The molecular characterization of NIL080 showed one additional introgression on the bottom of chromosome 2 (IND2-3976, SL4.0ch02:37,775,679) and one at the top of chromosome 12 (IND12-0379, SL4.0ch12:3,831,421).

In the breeding process, a good recovery of the cultivated genome was reached after three and four rounds of backcrosses and one round of self-pollination in the development of NIL115 and NIL080, respectively. This percentage of cultivated genome recovered was calculated based on the detected introgressions. NIL115 recovered 91.76%, and NIL080 recovered 96.07%.

### 2.3. Ripening-Related Transcript Polymorphism

A high percentage of ripening-related polymorphism between mature green (MG) and red ripe (RR) was revealed by cDNA-AFLP, i.e., 87.27% and 80.80% for NIL115 and NIL080, respectively (Figure 3). For NIL115, the highest number of exclusive transcript-derived fragments (TDF) was found in the MG stage, while for NIL080, it was found in the RR stage. In NIL115, one TDF that showed polymorphism between MG and RR was selected and sequenced. This TDF has sequence homology with a percent of identity of 88%, with a gene encoding an APETALA2 ethylene response transcription factor (AP2/ERF) that is located on the bottom of chromosome 5 (*Solyc05g051380*).

### 2.4. Differentially Expressed Ripening Proteins

From the quantitative proteomic analysis of pericarp proteins, a total of 1760 and 1563 proteins were identified in the NILs and the parental genotypes, respectively. Pairwise comparisons were made between MG and RR stages for each genotype (Appendix A). The presence of a DEP was considered when a *p*-value < 0.05 and Log_2_ fold change was above 1 or below −1 (Figure 4). The comparison between MG and RR stages for CAI showed 117 DEPs (8%) and 342 DEPs (22%) for LA0722. The results for NIL115 show 111 DEPs (6%), while for NIL080, 57 DEPs (3%) were identified (Figure 4).

Increased abundant proteins were majorly obtained at the RR stage in NIL115 (70 DEPs), but they were more often at the MG stage in NIL080 (41 DEPs). This difference was also observed in the parental genotypes. LA0722 presented a higher number of increased abundant proteins in the RR stage such as NIL115, while CAI showed the same behavior as NIL080 (Figure 4).

The gene ontology (GO) enrichment analysis highlighted overrepresentation of cytosol and plasma membrane GO categories (Appendix A). Chloroplastic proteins were also abundant. The molecular functions of the proteins were concentrated on binding and protein homodimerization activity in both NILs. Most of the DEPs focused on response biological processes.

The GO analysis also allowed us to identify proteins involved in ripening processes (Table 1). Six DEPs with an effect on tomato ripening were identified in NIL115, and four in NIL080. Among these proteins, it was analyzed whether their regulation was increased in MG, in RR, or was invariant in each NIL, and compared with the regulation pattern observed in the parents (Figure 5).

## 3. Discussion

Several NILs from our collection displayed long SL and reduced fruit softening provided by introgressions of the wild accession LA0722 from *S. pimpinellifolium* in a cultivated background, with NIL115 and NIL080 showing the most significant improvements (Figure 1). The increased SL was expected since this trait underwent selection pressure during the development of the NIL collection [32]. The usefulness of LA0722 as a source of genes to extend fruit SL without undesirable quality effects was previously shown [28,30,31]. Fruit F is another important attribute in fresh market tomatoes. The reduced fruit softening observed in these NILs is also beneficial in reducing postharvest waste. Furthermore, the recovery of cultivated fruit size and fleshiness is a positive factor in consumer preferences and can be attributed to the backcrossing process. These NILs provide valuable genetic resources for further studies on the genetic basis of fruit ripening and the components that define these traits.

Our NIL collection was developed following a marked-assisted selection with a set of 28 SSRs [32]. In the present study, the molecular characterization with 89 markers revealed that the wild SSR introgressions were not unique in NIL115 and NIL080 (Figure 2). Non-target segments in the development of NILs were previously reported by several authors [47,48], and massive genotyping in lines obtained using genetic maps of intermediate density showed that genetic background uniformity is often lower than expected [49,50]. Barrantes et al. [51] demonstrated that the inclusion of high-performance genotyping technologies in the early stages of a NIL breeding plan can guarantee the integrity of the introgression fragments. This approach can optimize the recovery of the recurrent background while reducing double recombinants that lead to additional introgressions. Despite the additional introgressions, a high recovery of the cultivated background could be obtained after three and four rounds of backcrossing in the two NILs under study.

Tomato is a model to study fleshy fruit ripening. We focused the study of the two NILs to gather additional information on the genetic bases underlying the transition from MG to RR stages. The transcript profiling showed that this transition involves many genes with differential expression (Figure 3). NIL115 displayed a higher number of exclusive TDFs in the MG stage, while NIL080 displayed a higher number in the RR stage. NIL115 also showed an increased SL. As this process begins in the MG stage, many ripening genes may be affected. In the MG stage, radical processes are taking place for fruit ripening, such as respiration, ethylene biosynthesis, and fruit softening [52]. At protein level, NIL115 had a higher percentage of polymorphism between MG and RR than NIL080 (Figure 4), but that percentage was significantly lower compared with the transcriptomic polymorphism. This could be attributed to a number of reasons, including the different sensitivity of each method and different mechanisms or points of regulation at several levels of the genetic information flow.

The transcript profiling analysis revealed differential expression of an AP2/ERF gene (*Solyc05g051380*) during fruit ripening in NIL115. AP2/ERF transcription factors belong to a protein superfamily conserved in the plant kingdom that is widely studied. The AP2/ERFs present various regulatory functions involved in processes such as the control of primary and secondary metabolism, growth and development, as well as responses to environmental stimuli [53]. Particularly, the *Solyc05g051380* gene was found to be a homolog of the *Arabidopsis* AINTEGUMENTA-LIKE6 (AIL6, At5g10510). It was demonstrated that AIL6 is involved in cell wall remodeling along with AINTEGUMENTA [54]. These transcription factors regulate the modifications to the cell wall polysaccharide pectin. AIL6 was associated with cell wall remodeling and pectin methylesterases (PME) inhibitor GO terms [54]. PMEs act to demethylesterify homogalacturonan (HG), the most abundant pectic polysaccharide in the cell wall. These demethylesterified HGs are substrates for enzymes that degrade pectin, such as polygalacturonase and PL, resulting in softening of the cell wall. Although the expression of Solyc05g051380 was shown to be induced in response to virus infection [55], its role in tomato ripening remains poorly understood. Within AP2/ERF superfamily, AP2a was identified as a major regulator of tomato fruit maturation [56]. Additionally, other AP2/ERFs were shown to be involved in this process [57]. Therefore, the differential expression of the transcription factor encoded by the gene *Solyc05g051380* might have a potential role in fruit ripening.

Proteins and their regulation pattern in the ripening process in both NILs and their parents were screened (Figure 5). We focused on the DEPs that play an important role in fruit ripening in the two NILs. NIL115 was characterized for increased fruit SL and F (Figure 1). Previous results in NIL115 show associations to ripening-related traits in chromosome 5 [32]. Adjacent to a wild introgressed region is located a PL gene (*Solyc05g014000*) that showed a differential expression between MG and RR stages in this NIL. PLs are well-studied proteins involved in cell wall remodeling and degradation that result in tomato fruit softening [9,39,58]. The *Solyc05g014000* gene presented a higher expression in the MG stage [2] in coincidence with our results (Figure 5, Table 1).

As a climacteric fruit, ethylene production is coupled to ripening and associated with fruit coloration and softening [40]. ERFs and ACO gene families are known to have an effect on fruit ripening. Our results show an increased protein expression of ACO3 (*Solyc09g089580*) and E4 (*Solyc03g111720*) in RR compared to MG (Table 1). The same expression pattern was observed in LA0722 (Figure 5). Moreover, in NIL115 near the genomic region of the E4 gene is located a homozygous wild introgression (Figure 2), indicating that it was possibly inherited from the wild parent. Previous studies revealed that ACO3 and E4 transcripts are accumulated when tomato ripening is initiated. They are expressed at low levels in MG, peak in breaker (B), and decline their expression in the RR stage; while for ripening mutants, including *rin* and *nor*, the high expression in the B stage was not observed [36,38,40]. In *Cnr* mutants, the expression peak of ACO3 was delayed when compared with the wild-type Ailsa Craig fruit [59].

A DEP with a role in climacteric ripening is a Cysteine synthase (*Solyc01g094790*). This protein was differentially expressed between MG and RR stages in NIL115 but was not in the parental genotypes. A recent study demonstrated that the expression levels of certain Cysteine synthase genes coincided with fruit ripening, suggesting their potential role in this process [33].

Phosphorylation plays an important role in the activation of AP2/ERFs [60]. We identified a MAPK that showed an upregulated expression during maturation codified on chromosome 9 (*Solyc09g091460*), which has a protein kinase domain of the Raf-like subfamily [41]. Preliminary STRING database research suggested that this enzyme might be involved in the regulation of the *Solyc05g051380* AP2/ERF (Appendix A). In *Arabidopsis*, ERFs were shown to act as substrates of MAPKs [61]. Phosphorylation appears to activate the transcriptional activity of ERFs in tomato as well as in rice and tobacco [53,62,63,64,65].

In NIL080, AGP1c (*Solyc05g018320*), an arabinogalactan, was identified as another DEP between MG and RR, and was invariant in the other genotypes (Figure 5). Arabinogalactan proteins (AGPs) are heavily glycosylated hydroxyproline-rich glycoproteins and their dynamic nature has an effect on fruit structure. Studies conducted by Leszczuk et al. [45] revealed that the decrease in AGPs content and pectin polysaccharides is associated with the remodeling of the cell wall and leads to fruit ripening and softening. Our results show an increased abundance of AGP1c in the RR stage (Table 1) indicating a possible effect of this protein in developing firmer fruits. In addition, in silico analysis allowed for identifying cis-acting elements of the promoter region of certain AGPs bound by specific transcription factors involved in tomato fruit ripening regulation [46].

During fruit ripening, there is also an efficient antioxidant system that protects from reactive oxygen species (ROS). ArcA2, a guanine nucleotide-binding protein (*Solyc03g119040*) was differentially expressed during NIL080 ripening. Wang et al. [42] performed a redox proteomic analysis in which ArcA2 was identified as a redox-sensitive protein during tomato ripening.

Another DEP identified in NIL080 was Pho1a (*Solyc03g065340*), an alpha-1,4 glucan phosphorylase. In concordance with our results, a previous study showed a high level gene expression in the MG stage [43]. This gene presented a correlation with starch degradation during fruit ripening. Li et al. [44] also revealed an effect of this gene in starch degradation and sugar accumulation. Furthermore, we identified a SSC QTL in the development of NIL080 [66].

There was one DEP of interest shared between MG and RR in both NILs (Figure 5); it corresponds to CHI17 (*Solyc02g082930*), an acidic endochitinase with a role in the defense against chitin-containing fungal pathogens. Eriksson et al. [67] found that chitinases were prevalent in Cnr fruits and might contribute to cell separation as well as to protecting the tissues from pathogen invasion. Additionally, several endochitinases with altered expression profiles were identified when comparing tomato ripening stages [68]. Only the NIL080 displayed a wild introgression near the region of CHI17; however, CHI17 was not differentially expressed in LA0722 or CAI (Figure 5). It suggests that interactions between wild introgressions and the cultivated background may result in this new DEP on both NILs.

Improvement of fruit quality and reduction in postharvest losses are two major challenges for tomato breeding. Thus far, these improvements were achieved either by gene edition or using hybrids with ripening mutants. In the first case, the consumer acceptance is still in discussion, while in the second, it is known that mutants on master transcription factors also have undesirable pleiotropic effects on fruit quality. In the present study, we demonstrated that the extension of fruit SL and the reduction in fruit softening could be achieved from classical marker-assisted breeding by modifying the expression of genes without the alteration of fruit quality.

## 4. Materials and Methods

### 4.1. Plant Material

The Argentinian cultivar Caimanta (CAI) of *S. lycopersicum* and the wild accession LA0722 of *S. pimpinellifolium* (LA0722) were used to obtain a collection of NILs [32]. Throughout the NIL development process, the phenotypic selection was based on SL and a marker-assisted selection was performed with a set of single sequence repeats (SSR) markers distributed throughout the tomato genome [32]. A collection of 22 NILs containing homozygous LA0722 wild introgressions in the cultivated background of CAI were obtained. From this collection, NIL115, a third backcross with one round of self-pollination (BC_3_S_1_), carries a wild homozygous introgression in the SSR115 locus (SL4.0ch05:2,779,337) associated with firmer fruits and longer SL. Furthermore, NIL080, a BC_4_S_1_, showed larger fruits with no differences in its cultivated parent and a QTL for SL. This introgression in the SSR080 locus (SL4.0ch11:2,309,947) in the heterozygous state delays fruit ripening. Based on these phenotypic characteristics, NIL115 and NIL080 were selected to further investigate the role of the LA0722 introgressions over fruit ripening (Appendix A).

### 4.2. Phenotyping

Trials were performed at the Estación Experimental José F. Villarino (33° 02’ S and 60° 53’ W, Facultad de Ciencias Agrarias, Universidad Nacional de Rosario, Argentina). Phenotyping was conducted on fruits collected from 10 plants of each NIL. Ten plants of the parental genotypes (CAI and LA0722) were used as controls. Plants were distributed following a completely random design in a greenhouse. Following methodologies described by Di Giacomo et al. [32], these 13 fruit quality traits were measured: diameter (D, cm), height (H, cm), shape (Sh, H/D ratio), weight (W, g), pericarp thickness (PT, cm), locule number (LN), shelf life (SL, days), fruit firmness (F), color indexes (a/b and L), soluble solids content (SSC, °Brix), and pH and titratable acidity (TA, g citric and malic acid/100 g of homogenized juice). Dunnett tests were performed to determined statistically pairwise differences (*p*-value < 0.05) between LA0722, NIL115, and NIL080 genotypes against the recurrent parent CAI.

### 4.3. Genomic Approach: Molecular Characterization

Genomic DNA was extracted from young leaves of six plants from NIL115 and 10 plants of NIL080 using a commercial kit (Wizard^®^ Genomic DNA Purification Kit from Promega). The extracted DNA was dissolved in buffer TE (10 mM Tris-Cl pH 8.00, 1 mM EDTA) and the final concentration was adjusted to 40 ηg/μL. A set of 89 molecular markers (Appendix A) were used for the molecular characterization of each NIL in order to determine the background recovery and additional wild introgressions [66,69].

### 4.4. Transcript Profiling: cDNA-AFLP

Pericarp tissue was collected from a single fruit of three different plants per NIL (three biological replicates) at two ripening stages: mature green (MG) and red ripe (RR). MG corresponds to the end of cellular expansion, when fruit growth stopped and fruits start to ripe, whereas RR corresponds to the mature fruit [52]. The total number of fruits per genotype was six. Pericarp samples were frozen in liquid nitrogen and stored at −80 °C.

Total RNA was extracted with TriPure Isolation Reagent according to the manufacturer’s instructions (Roche, Basel, Switzerland). Briefly, 500 mg of grounded pericarp was mixed with 1 mL of TriPure Isolation Reagent. A volume of chloroform was added and the aqueous phase containing RNA was separated. RNA was finally precipitated with 0.5 mL of isopropanol.

The cDNA synthesis to obtain cDNA-AFLP profiles was performed according to Pereira da Costa et al. [70]. The first and second cDNA strand were obtained from 1 μg of total RNA. Digestion was performed using restriction enzymes *ApoI* and *MseI* [71]. Adapter sequences, ligation and amplification conditions were carried out following the protocol proposed by [72]. A pre-amplification was performed using primer sequences with no selective base at the end (+0). For selective amplification, primer sequences ended with one selective base (+1) in four combinations were used (Appendix A). Selective amplification products were separated on 6% *w*/*v* polyacrylamide gels running at 50 W for 3 h and visualized by silver staining (Silver Sequence™ Staining Reagents, Promega, Madison, WI, USA).

The identification of transcript-derived fragments (TDF) was performed on the basis of their differential expression pattern. The presence or absence of a TDF was considered when two or more biological replicates presented a band or not, respectively. The two ripening stages in each NIL were compared. The total number of TDFs and polymorphic TDFs per ripening stage were calculated.

TDFs were cut, eluted, and re-amplified to be sequenced based on the differential expression observed between ripening stages. In NIL115, one TDF that showed polymorphism between MG and RR was sequenced, but in NIL080, the ripening-polymorphic TDFs were not possible to re-amplify. Sequencing was carried out by Macrogen Inc. (Seoul, Republic of Korea). Sequence analysis was performed through the BLASTn tool (www.solgenomics.net accessed on 5 December 2019) using the Tomato Genome CDS (ITAG release 4.0) and *S. pimpinellifolium* CDS LA1589 databases.

### 4.5. Proteomic Approach: Label Free Quantification

The same tissue described for the transcript profiling was used for this experiment (Section 4.4). Here, we also included the parental genotypes (CAI and LA0722) for comparisons. Proteins were extracted from 1 g of pericarp per genotype–ripening stage combination according to Wu et al. [73]. Three technical replicates were used in this experiment. The protein pellets were dissolved in 60 μL of 6 M urea and 1% *w*/*v* CHAPS. Total protein concentration was quantified by absorbance at 280 nm and gel densitometry after the extraction procedure. A total number of 24 samples were sent to the CEQUIBIEM Proteomics Facility (Universidad de Buenos Aires, Buenos Aires, Argentina) for protein digestion and mass spectrometry (MS) analysis. Each sample was reduced with 20 mM DTT for 45 min at 56 °C, alkylated with 50 mM Iodoacetamide for 45 min in the dark, and digested with trypsin overnight. Extraction of peptides was performed with acetonitrile and salt cleaning was carried out through Zip-Tip C18 (Merck, Darmstadt, Germany). Desalted peptides were analyzed by nano-high performance liquid chromatography (EASY-nLC 1000, Thermo Scientific, Waltham, MA, USA) coupled to an Orbitrap technology mass spectrometer (Q-Exactive, high collision dissociation cell and Orbitrap analyzer Thermo Scientific, Waltham, MA, USA). Peptide ionization was performed by electrospray (Easy Spray, Thermo Scientific, Waltham, MA, USA) at 2.5 kV.

The obtained spectra were analyzed using Proteome Discoverer 2.2 software (Thermo Scientific, Waltham, MA, USA). The search was performed using 10 ppm precursor ion mass tolerance and 0.05 Da fragmentation mass tolerance with a 1% false discovery rate. Tryptic cleavage was then selected, and up to two missed cleavages were allowed. Oxidation on methionine was used as dynamic modification and carbamidomethylation on cysteine was used as static modification. For protein identification, searches were conducted against the *Solanum lycopersicum* cv. Heinz 1706 database (UniProt, proteome reference: UP000004994). The MS proteomics data were deposited to the ProteomeXchange Consortium via the PRIDE [74] partner repository with the dataset identifier PXD036132.

Protein intensities were log transformed. The imputation of Log_2_ intensity values were carried out using a downshifted normal distribution with a width of 0.3 and downshift of 1.8 for each sample. The relative abundance of peptides across ripening stages was compared using R software (version 3.6.3). Briefly, the empirical Bayes analysis pipeline of the Bioconductor limma package was used to obtain the differential expressed proteins (DEPs) between the two ripening stages (MG versus RR) in each genotype [75]. A *p*-value < 0.05 and values of Log_2_ fold change above 1 or below −1 were set to determine significantly increased abundant proteins in the MG or RR stages. A gene ontology (GO) term enrichment analysis was performed using agriGO v2.0 software [76]. Protein–protein interaction networks were analyzed using the STRING database [77].

### 4.6. Statistical Analysis

Statistical data analysis was performed using R software (version 3.6.3). A multiple comparison Dunnett test was applied to identify the statistically pairwise differences. This many-to-one comparison allowed for detecting the phenotyping differences between the NILs and LA0722 in comparison to CAI. A *p*-value < 0.05 was set. On the other hand, a statistical comparison test was performed to identify the DEPs between the two ripening stages. A linear model fit from *lmFit* of the limma package was used; it computes moderated t-statistics and the log odds of differential expression by empirical Bayes moderation of the standard errors towards a global value. To identify a DEP between MG and RR stages, a *p*-value < 0.05 was considered.

## 5. Conclusions

Data integration allowed us to unravel genes underlying tomato ripening in the NILs. These include genes associated to cell wall remodeling such as AP2/ERF, PL, and AGP1c; others involved in ethylene pathways are E4 and ACO3; and also others are involved in biotic and abiotic stress (CHI17 and ArcA2) and metabolism pathways (Cysteine synthase and Pho1a)—all with a role in climacteric ripening. Comparing with the parental lines, some ripening proteins displayed the same regulation as one of the parental genotypes. NIL115, E4, ACO3, and MAPK were found to be differentially expressed in the wild parent LA0722 but not in the cultivated parent CAI. As a result, these three DEPs were selected from the wild parent in the breeding process. On the other hand, several proteins showed a unique differential expression that was not observed in either of the parental genotypes. Therefore, combinations and interactions between donor wild alleles and the receptor background are broadening the phenotypic variability and influencing fruit ripening in the NILs. Here, we provided valuable insights into the effect of several genes underlying tomato ripening with a positive impact. To build upon these results, future research should focus on further investigating and functionally validating these genes. Additionally, the application of high-throughput techniques for genome and transcriptome sequencing holds great potential. By utilizing these techniques, we will be able to not only detect new wild introgressions but also achieve a more precise molecular delimitation of known ones and discover novel ripening-related genes with differential expression. By integrating different omics approaches, we unraveled key ripening-related genes, enhancing our understanding of tomato ripening and providing opportunities for tomato assisted breeding.

## Figures and Tables

**Figure 1 plants-12-02812-f001:**
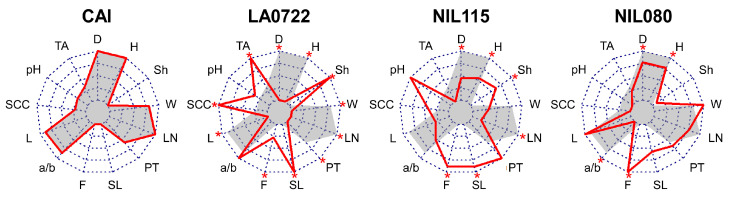
Phenotyping based on 13 fruit quality traits for the parental genotypes (CAI and LA0722) and the derived NILs (NIL115 and NIL080). Circles correspond to a 0–1 scale of data normalized to the minimum or maximum value for each trait. Color background indicates CAI values, and bold lines indicate the values of each genotype. Asterisks show significant differences with CAI (Dunnett test, *p*-value < 0.05). D, diameter; H, height; Sh, shape; W, weight; LN, locule number; PT, pericarp thickness; SL, shelf life; F, firmness; a/b, color index a/b; L, color index L; SSC, soluble solids content; and TA, titratable acidity. Raw data in Appendix A.

**Figure 2 plants-12-02812-f002:**
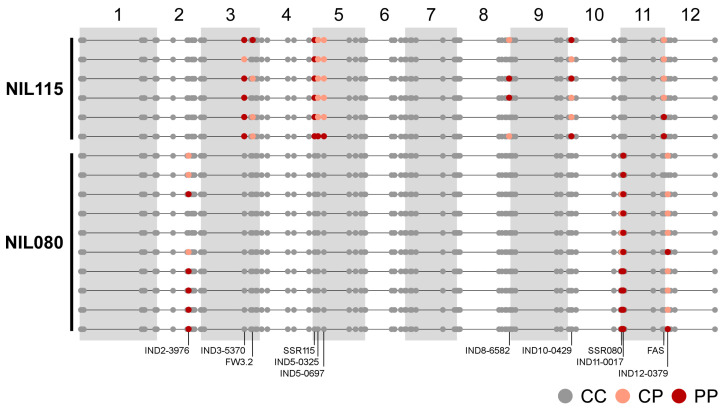
Schematic representation of the molecular characterization based on 89 molecular markers for NIL115 and NIL080. CC dots correspond to homozygous loci as the recurrent parent *S. lycopersicum* cv. Caimanta; PP dots to homozygous wild introgressions from LA0722 of *S. pimpinellifolium;* and CP dots to heterozygous state. Chromosome numbers are shown on the top and markers names are at the bottom. Positions are relative to the tomato genome sequence reference version SL4.0. References in Appendix A.

**Figure 3 plants-12-02812-f003:**
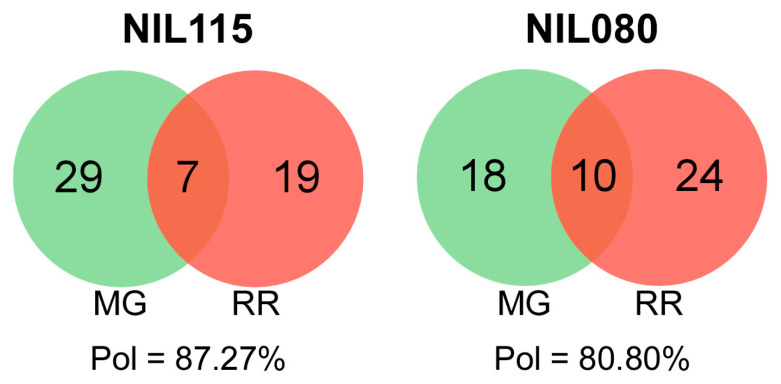
Venn diagrams of total and exclusive transcript-derived fragments (TDFs) for each ripening stage of the lines NIL115 and NIL080. TDFs were obtained by AFLP-based transcript profiling (cDNA-AFLP) with four specific primer combinations. Pol represents percentages of polymorphism between mature green (MG) and red ripe (RR) stages.

**Figure 4 plants-12-02812-f004:**
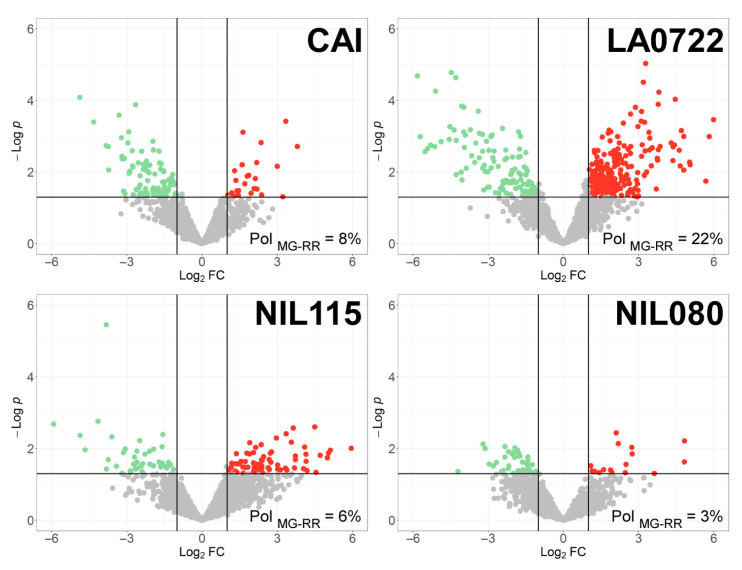
Volcano plots. Differentially expressed proteins between mature green (MG) and red ripe (RR) for the original parental genotypes Caimanta (CAI) of *S. lycopersicum* and LA0722 (LA0722) of *S. pimpinellifolium* and two derived NILs from its crossing (NIL115 and NIL080). Green dots show proteins with increased abundance in the MG stage, red dots show proteins with increased abundance in the RR stage, and grey dots show invariant proteins between the two ripening stages. Pol is the percentages of polymorphism between mature green (MG) and red ripe (RR) stages; and FC is the fold change.

**Figure 5 plants-12-02812-f005:**
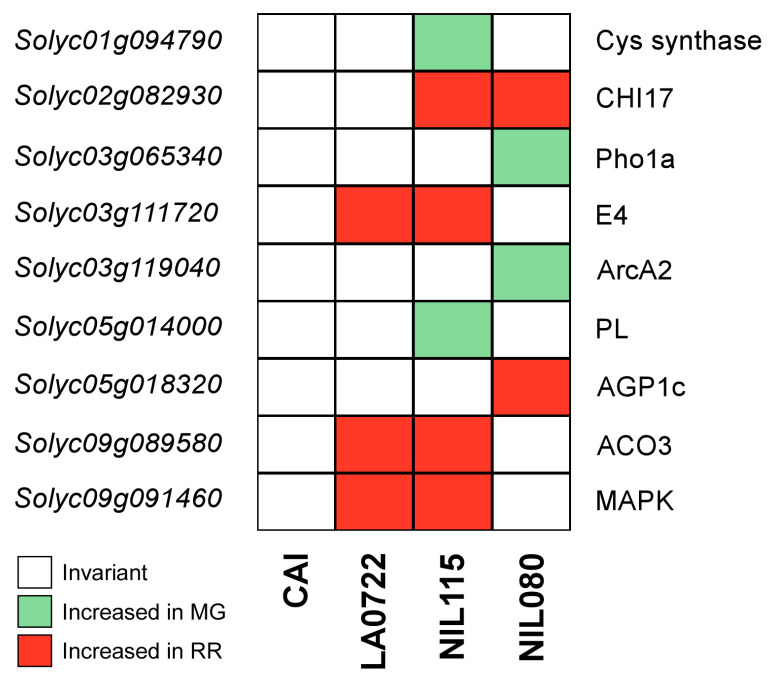
Regulation of the significantly differential expressed ripening-related proteins (DEPs) between mature green (MG) and red ripe (RR) in NIL115, NIL080, and their parents (CAI and LA0722). The DEPs are indicated in color blocks (*p*-value < 0.05); green ones for the DEPs increased in MG (Log_2_ fold change below −1) and red ones for the DEPs increased in RR (Log_2_ fold change above 1). White blocks indicate proteins with invariant expression between the two ripening stages.

**Table 1 plants-12-02812-t001:** Differentially expressed proteins (DEPs) between mature green (MG) and red ripe (RR) in the lines NIL115 and NIL080 with a role in fruit ripening.

NIL	Solyc ID	UniProt ID	Protein Name	Function	Increased Abundance	References
NIL115	*Solyc01g094790*	A0A3Q7F5F8	Cys synthase	Cysteine synthase	MG	Liu et al., 2019 [33]
	*Solyc02g082930*	Q05540	CHI17	Acidic endochitinase	RR	Cao and Tan, 2019 [34] Celik et al., 2023 [35]
	*Solyc03g111720*	P54153	E4	Peptide methionine sulfoxide reductase	RR	Martel et al., 2011 [36] Li et al., 2019 [37] Gao et al., 2020 [38]
	*Solyc05g014000*	A0A3Q7GFD0	PL	Pectate lyase	MG	Seymour et al., 2013 [39]
	*Solyc09g089580*	P10967	ACO3	ACC oxidase	RR	Barry and Giovannoni, 2007 [40] Martel et al., 2011 [36] Li et al., 2019 [37] Gao et al., 2020 [38]
	*Solyc09g091460*	A0A3Q7I9A3	MAPK	Protein kinase of the Raf-like subfamily	RR	Iftikhar et al. 2017 [41]
NIL080	*Solyc02g082930*	Q05540	CHI17	Acidic endochitinase	RR	Cao and Tan, 2019 [34] Celik et al., 2023 [35]
	*Solyc03g119040*	A0A3Q7GJ89	ArcA2	Redox-sensitive guanine nucleotide-binding protein	MG	Wang et al., 2021 [42]
	*Solyc03g065340*	A0A3Q7G8G0	Pho1a	Alpha-1,4 glucan phosphorylase	MG	Slugina et al., 2019 [43] Li et al., 2021 [44]
	*Solyc05g018320*	A0A3Q7GJJ9	AGP1c	Arabinogalactan	RR	Leszczuk et al., 2018 [45] Leszczuk et al., 2020 [46]

## Data Availability

Data supporting the findings of this work are available within this published article and the Electronic Appendix A. The proteomics data generated in this study have been deposited to the ProteomeXchange Consortium via the PRIDE partner repository with the dataset identifier PXD036132.

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
