# Peer review of "An Integrative Transcriptomics and Proteomics Approach to Identify Putative Genes Underlying Fruit Ripening in Tomato near Isogenic Lines with Long Shelf Life"

_plants, 2023, doi:10.3390/plants12152812_

Round 1
Reviewer 1 Report
Manuscript "An integrative omics approach to identify putative genes underlying fruit ripening in tomato near isogenic lines with long shelf life" is interesting.
Authors gathered additional evidence of tomato fruit ripening by assessing two long SL NILs with S. pimpinellifolium wild introgressions. Authors performed a comprehensive analysis at different molecular levels including genomic, transcriptomic and proteomic approaches.
Minor corrections:
Figure 4: The decimal logarithm does not require the number "10". This should be corrected.
Figure 5: Complete with a criterion for the significance of the difference. Specify the method used and the critical value.
The Material and Methods chapter should be supplemented with a subsection titled "Statistical Analysis," including a description of all statistical methods used in the manuscript.
Paper needs major revision.
Reviewer 2 Report
The authors described the integrative omics approach to identify putative genes un- derlying fruit ripening in tomato near isogenic lines with long shelf life
The experimental design is quite straight-forward, and it is well-prepared to understand.
It was easy to follow the mainstream of the study.
The following points should be corrected for publication.
The title should be corrected to reflect the specific techniques (transcriptomics and proteomics) they employed.
In conclusion section;
The limitation of the study should be added.
In addition, further studies to be performed also should be added.
Round 2
Reviewer 1 Report
Authors have incorporated all the suggestions, accordingly. I recommend this article to publish in current version.
Author Response
Thanks so much for giving us the possibility to revise and improve the manuscript. We truly appreciate your comments.